

# Generative AI and future education: a review, theoretical validation, and authors' perspective on challenges and solutions

Wali Khan Monib[1], Atika Qazi[1], Rosyzie Anna Apong[2], Mohammad Tazli Azizan[1], Liyanage De Silva[2,3] and Hayati Yassin[3]

[1] Centre for Lifelong Learning, Universiti Brunei Darussalam, Gadong, Brunei Darussalam
[2] School of Digital Science, Universiti Brunei Darussalam, Gadong, Brunei Darussalam
[3] Faculty of Integrated Technologies, Universiti Brunei Darussalam, Gadong, Brunei Darussalam

## ABSTRACT

Generative AI (Gen AI), exemplified by ChatGPT, has witnessed a remarkable surge in popularity recently. This cutting-edge technology demonstrates an exceptional ability to produce human-like responses and engage in natural language conversations guided by context-appropriate prompts. However, its integration into education has become a subject of ongoing debate. This review examines the challenges of using Gen AI like ChatGPT in education and offers effective strategies. To retrieve relevant literature, a search of reputable databases was conducted, resulting in the inclusion of twenty-two publications. Using Atlas.ti, the analysis reflected six primary challenges with plagiarism as the most prevalent issue, closely followed by responsibility and accountability challenges. Concerns were also raised about privacy, data protection, safety, and security risks, as well as discrimination and bias. Additionally, there were challenges about the loss of soft skills and the risks of the digital divide. To address these challenges, a number of strategies were identified and subjected to critical evaluation to assess their practicality. Most of them were practical and align with the ethical and pedagogical theories. Within the prevalent concepts, "ChatGPT" emerged as the most frequent one, followed by "AI," "student," "research," and "education," highlighting a growing trend in educational discourse. Moreover, close collaboration was evident among the leading countries, all forming a single cluster, led by the United States. This comprehensive review provides implications, recommendations, and future prospects concerning the use of generative AI in education.

# INTRODUCTION

Artificial Intelligence (AI) refers to the field where machines or computer programs are designed to perform tasks that typically require human intellect, such as language processing, learning, problem-solving, and decision-making (*Dalalah & Dalalah, 2023*). Within AI, Gen AI constitutes a subset designed to produce new content, such as text, images, audio, or other data formats, often in a creative or human-like fashion. At the forefront of AI research and development stands OpenAI, a research company dedicated

Corresponding author
Atika Qazi, atika.qazi@ubd.edu.bn

to advancing AI technology (*Yilmaz & Yilmaz, 2023*). Among OpenAI notable achievements is ChatGPT (*Yilmaz & Yilmaz, 2023*), a prominent member of the generative pre-training transformer (GPT) model family and the largest publicly accessible language model (*Dave, Athaluri & Singh, 2023*) through various web browsers, including Safari, Firefox, and Chrome, on desktop computers and mobile devices (*Barrot, 2023*).

Since its launch in November 2022, ChatGPT has quickly gained immense popularity and become a prominent chatbot on the Internet (*Barrot, 2023*). Both ChatGPT and GPT-3 are notable examples of natural language processing (NLP) applications (*Howell & Potgieter, 2023*). *Barrot (2023)* highlights the AI tool stays current and well-informed through ongoing updates and *corpus* expansions, with the latest iteration, GPT-4, introduced in March 2023, capable of processing 32,000 tokens concurrently.

ChatGPT utilizes advanced computing techniques and extensive data to establish connections between words and ideas, enabling it to comprehend prompts within their context. Powered by a robust large language model (LLM), this tool generates "human-like" responses that closely resemble human language on diverse topics (*Barrot, 2023*; *Bewersdorff et al., 2023*; *Currie, 2023*). ChatGPT relies on an extensive collection of statistical patterns and associations, rather than a conventional database, to produce context-specific and logically relevant responses (*Dalalah & Dalalah, 2023*). It utilizes unsupervised pre-training and supervised fine-tuning techniques to generate human-like responses to queries on various topics (*Dwivedi et al., 2023*). Additionally, sophisticated chatbots utilize algorithms and predictive text to generate new content in response to users' prompts (*Sweeney, 2023*). The algorithm used in ChatGPT is evolutionary, meaning that the training data consists of large volumes of data with high velocity, heterogeneity, and variability characteristics (*Dwivedi et al., 2023*). They further state that the adoption of chatbots by humans initially aimed at automation, but it quickly evolved to the augmentation of human tasks and the formation of hybrid teams, where humans and machines closely collaborate to enhance the execution of relatively simple tasks that made a significant and historically transformative development. However, its emergence and transformative nature have caused intense debate on the implications for educational practices, raising questions about the role of AI in shaping the future of teaching and learning. Acknowledging the rapid evolution of technology in education, UNESCO has issued a call for the responsible utilization of technology, emphasizing the need for governance and regulation to adapt to the rapid pace of technological change and prioritize quality education for all (*Azoulay, 2023*). In this comprehensive review, we explore Gen AI like ChatGPT in education, particularly the challenges posed by such tools, and how to turn these challenges into opportunities for learning enhancement.

## Gen AI technologies

In this section, we provide a concise overview of state-of-the-art Gen AI technologies employed for various purposes.

### GPT

GPT, the Generative Pre-Trained Transformer series, developed by OpenAI, consists of neural language models proficient in sequential text generation (*Shaik Vadla, Suresh & Viswanathan, 2024*). GPT was designed at predicting and completing human-generated text, finding utility in diverse applications like machine learning and text prediction (*Roumeliotis & Tselikas, 2023*). The GPT series consists of models pre-trained on a blend of five datasets, including Common Crawl, WebText2, Books1, Books2, and Wikipedia, boasting an impressive parameter count of 175 billion (*Shaik Vadla, Suresh & Viswanathan, 2024*). The development of GPT laid the foundation for ChatGPT (*Roumeliotis & Tselikas, 2023*). According to *Yilmaz & Yilmaz (2023)*, five versions of ChatGPT have been introduced to date: GPT-1, released in 2018 with 117 million parameters, performed relatively poorly compared to later models; GPT-2, released in 2019, improved significantly with 1.5 billion parameters and better language generation; GPT-3, released in 2020, further enhanced language generation with 175 billion parameters and a larger dataset. GPT-3.5 served as an intermediate model before the introduction of GPT-4, which exhibits broad knowledge and improved problem-solving capabilities for accurate strategies. Earlier versions of GPT were less advanced and did not fully represent the current capabilities of the model (*Dwivedi et al., 2023*).

### BART

Bidirectional and Auto-Regressive Transformers (BART), developed by Facebook AI, is an NLP pre-trained model that combines bidirectional and auto-regressive Transformer (*Lewis et al., 2019*). It is a modified BERT with an emphasis on natural text generation (*Alokla et al., 2022*) and generation tasks, such as text summarization, text generation, and language translation (*Radanliev, 2024*). It combines an encoder and decoder, trained with extensive unlabeled data, to achieve state-of-the-art results (*Venkataramana, Srividya & Cristin, 2022*). The encoder role is to map the input sequence into an intermediate representation, while the decoder converts the intermediate representation back into the original input space (*Liu, Ju & Wang, 2024*). This architecture facilitates the model ability to grasp intricate relationships within the inputs. Its performance on various generation tasks positions it as a valuable tool, particularly for tasks requiring both understanding and generation of natural language (*Chen et al., 2020*).

### LLaMA

Large Language Model Meta AI (LLaMA), unveiled by Meta AI in February 2023, represents a significant milestone in large language model development. Built on the transformer architecture (*Radanliev, 2024*), LLaMA promises enhanced capabilities in language understanding and generation. It is explicitly designed for research purposes, specifically to evaluate and compare existing language models across various natural language processing tasks (*Shaik Vadla, Suresh & Viswanathan, 2024*). It helps researchers advance their work, having conversation, and summarizing written materials (*Belagatti, 2023*). It is trained across a range of parameters from 7 billion to 65 billion that

encompasses diverse evaluation metrics and tasks (*Shaik Vadla, Suresh & Viswanathan, 2024*), offering a holistic approach to assess and compare language model performance in NLP (*Oralbekova et al., 2023*).

### T5

Another tool is T5 (Text-To-Text Transfer Transformer), an encoder-decoder transformer (*Raffel et al., 2020*) utilized for transfer learning to produce a unified framework for multiple NLP tasks (*Al-Qaraghuli & Jaafar, 2024*). T5 was introduced by Google and functions on two fronts: the encoder comprehends input text, discerning its semantics and word relationships, while the decoder formulates responses, crafting new content such as summaries, translations, or dataset labels (*Shaik Vadla, Suresh & Viswanathan, 2024*). It has been used for tasks such as translation, text generation (*Chomphooyod et al., 2023*), and summarization (*Venkataramana, Srividya & Cristin, 2022*). It offers varied sizes, from small to T5-11b, for adaptable pre-training suited to diverse NLP tasks, making it versatile for text-based applications (*Oralbekova et al., 2023*).

### BARD

Bayesian Adaptive Representations for Dialogue (BARD) is a substantial language model primarily targeting computational efficiency within transformer-based NLP models (*Shaik Vadla, Suresh & Viswanathan, 2024*). Google BARD, launched on March 21, 2023, emerges as a direct competitor to ChatGPT (*Moons & Van Bulck, 2023*). Developed with similar training method (*Bridgelall, 2024*), it has the same aims as ChatGPT (*Moons & Van Bulck, 2023*). The main distinguishing feature lies in its foundation on the LaMDA family of large language models (*Moons & Van Bulck, 2023*). BARD offers cost-effectiveness, adaptability, and unique features like internet access for real-time information, giving it a competitive edge over ChatGPT in dialogue-based language models (*Shaik Vadla, Suresh & Viswanathan, 2024*). This study reviews the challenges posed by Generative AI, with an emphasis on ChatGPT, within the context of education.

## Gen AI usage in education

The integration of Gen AI in education is remarkably swift, surpassing the lengthy processes for validating new textbooks (*Giannini, 2023*). Recent advancements in Gen AI, exemplified by ChatGPT, have demonstrated the ability to generate highly convincing output closely resembling human-authored content (*Currie, 2023*; *Howell & Potgieter, 2023*; *Májovský et al., 2023*). For example, as part of the GPT series, ChatGPT-3, has demonstrated its ability to autonomously generate credible academic article with minimal human input and GPT-4 can handle visual queries and possesses an impressive 100 trillion parameters (*Barrot, 2023*). By leveraging extensive text data, ChatGPT captures the nuances of human language, facilitating contextually relevant responses across diverse prompts and streamlining research article writing by providing direct answers and eliminating manual article searches (*Dave, Athaluri & Singh, 2023*). Such Gen AI tools excel at tasks like essay generation, intricate question answering, pronoun-noun agreement, sentiment analysis (*Sezgin, Sirrianni & Linwood, 2022*), text summarization, code generation, and even story creation (*Howell & Potgieter, 2023*). However, the

pervasive use of such tools raises concerns regarding the potential for learners to become overly dependent on it, resulting in a decline in essential soft skills. Moreover, it is essential to acknowledge that Gen AI tools may generate fraudulent content, which can not only mislead learners but also foster incorrect learning. For example, a study by *Májovský et al. (2023)* revealed that the AI language model can produce fraudulent articles that closely mimic genuine scientific articles, exhibiting convincing word usage, sentence structure, and overall composition. Additionally, tools like ChatGPT, with access to open-access data, can inadvertently introduce biases and discrimination in the output, potentially impacting the learning process. They analyze large corpora from diverse sources such as Wikipedia and Reddit to generate grammatically accurate content (*Sweeney, 2023*). Therefore, it is important to acknowledge that, in the absence of human oversight, there is a risk of misleading or inaccurate content (*Stokel-Walker & Van Noorden, 2023*). It is noteworthy that tools like ChatGPT recognize the dual impact of technology in education, akin to the controversy surrounding calculators in the 1970s (*Dwivedi et al., 2023*). They further state that it would be illogical to prohibit the use of calculators for students. Therefore, it is crucial for education to be finely attuned to address AI risks, considering both known and emerging concerns (*Giannini, 2023*). This comprehensive review aims to answer the following research question:

**RQ1:** What are the challenges of using Gen AI like ChatGPT in education?

**RQ2:** How can the challenges posed by Gen AI like ChatGPT in education be addressed?

## Rationale and intended audience

This review provides crucial contributions to academic discourse by comprehensively exploring Gen AI in education. First, the review aims to synthesize existing knowledge and research in this area, providing a cohesive overview of the current state of Gen AI with a focused on ChatGPT research. By consolidating information, it enables researchers and scholars to gain a holistic understanding of challenges, reflections, and solutions related to integrating Gen AI in education. Furthermore, the synthesized solutions or strategies are critically evaluated in light of relevant theories and frameworks, offering insights for future Gen AI usage in educational settings. This aspect of the review is particularly valuable for academics and researchers seeking to establish a solid theoretical foundation for their work in the intersection of AI and education. Theoretical validation contributes to the establishment of a robust framework that guides future research and development in this dynamic field. Additionally, the review provides authors' perspective on the solutions or strategies pertaining to the implementation of Gen AI in education. This perspective can be especially beneficial for educators, educational technologists, and policymakers, offering them valuable insights into potential practicality of solutions or strategies for effectively incorporating Gen AI into educational practices. This includes an assessment of the feasibility of applying the derived strategies in educational settings.

The intended audience for this comprehensive review encompasses a wide spectrum. It caters to researchers in Gen AI and education who seek a nuanced understanding of current implications of Gen AI in education. Educators and educational technologists will

find the review valuable as it addresses challenges, and provide practical solutions of integrating Gen AI in education. Furthermore, policymakers and administrators can leverage the insights from the review to make informed decisions regarding the adoption of AI technologies such as ChatGPT in educational institutions. Lastly, learners interested in AI-driven learning will find the review valuable, offering insights into effectively using emerging technologies like ChatGPT to mitigate negative consequences.

# METHODOLOGY

## Literature search

In this comprehensive review, the literature search covered multiple reputable databases, including Emerald, Scopus, Sage, Springer, Taylor & Francis, IEEE, and ERIC. The search was limited to the title, abstract, and keywords of the publications. A comprehensive set of keywords was employed. The Boolean operators included "ChatGPT" AND ("education" OR "educational" OR "learning" OR "teaching" OR "classroom" OR "school" OR "university" OR "student" OR "learner" OR "teacher" OR "instructor" OR "academic" OR "teaching" OR "instruction") AND ("challenges" OR "limitations" OR "obstacles" OR "cons" OR "drawbacks" OR "ethical considerations" OR "privacy" OR "data security" OR "bias" OR "fairness" OR "ethical implications" OR "legal implications" OR "equity" OR "accessibility"). However, the exclusion of databases beyond those listed may result in overlooking relevant literature. The search of specific keywords and Boolean operators may inadvertently exclude related studies. Additionally, the focus on challenges have limited the scope of the study, excluding studies exploring other aspects other than challenges.

## Inclusion and exclusion

We included relevant literature written in English and published between 2022 and 2023 that specifically mentioned challenges, limitations, threats, and/or concerns related to Gen AI, ChatGPT usage in educational settings. To ensure a comprehensive review and capture all relevant insights, the search was not restricted to a specific document type. In addition, the UNESCO website was explored because it provides reliable, global, and up-to-date educational content, policy resources, insights into current trends, and guidance on educational technology. We excluded literature that did not focus on the challenges of ChatGPT in education and literature published in languages other than English. As a result, the final selection comprised ($n$ = 22) publications including one notes, and two editorials. The evident quality appraisal of the selected publications is indicated by their retrieval from reputable sources as outlined in the literature search section, aligning with the comprehensive review approach.

## Analysis

In this review, we utilized ATLAS.ti, a robust qualitative data analysis software, to efficiently manage and analyze the selected publications. We initiated by importing the chosen documents into the software, facilitating efficient coding and categorization. Thematic analysis, a qualitative method for non-numerical data, was employed, involving

the identification of codes from the data which serve as categories for analysis (*Roberts, Dowell & Nie, 2019*). Coding approach include inductive, starting from data to identify meaning, and deductive, where pre-existing ideas or concepts are applied to the data (*Braun et al., 2019*). We followed the deductive analysis, where pre-existing ideas or concepts were applied based on prior knowledge and literature. This approach involves predetermining (at least some of) the codes before commencing the analysis (*Wæraas, 2022*). This allowed us to effectively analyze and interpret the selected literature.

## RESULT AND DISCUSSION

This section reports the results and discussion of the selected literature.

### Concepts in the analyzed literature

The analysis in Fig. 1 reveals core concepts within the reviewed literature. Notably, "ChatGPT" emerged 2,520 times, and "AI" 1,767 times, indicating a significant trend in education. "Student" appeared 1,705 times, while "research" and "education" were mentioned 920 and 873 times, respectively. These frequencies underscore the central concepts of "ChatGPT" and its emphasis on AI, students, research, education, among others. Education needs to be finely attuned to the risks of AI, both the known risks and those only just coming into view (*Giannini, 2023*).

### Countries collaboration

Figure 2 illustrates an overlay visualization of the collaborative network among countries in research related to Gen AI, such as ChatGPT. The analysis involved co-authorship, with the unit of analysis being countries with a minimum of two documents, utilizing a full counting method. Of the 30 countries, eight met the threshold in the selected publications sourced from Scopus. In the visualization, circle size corresponds to document frequency, lines depict collaborations, and the thickness of the lines indicates the frequency of collaboration. As observed, close collaboration is evident among countries such as the United States, Australia, Germany, and India, Hong Kong, Netherlands, Oman, and the United Kingdom. Notably, all are clustered together in cluster 1, highlighting a significant gap due to the absence of collaboration with other regions.

### RQ1. What are the challenges of using Gen AI like ChatGPT in education context?

Deductive thematic analysis was conducted, resulting in six overarching challenges regarding the usage of Gen AI in education, see Fig. 3. Foremost among these concerns is the category of plagiarism, followed by responsibility and accountability. Concerns were also raised about privacy, data protection, safety, and security risks, as well as discrimination and bias. Additionally, there were concerns about the loss of soft skills and the risks of the digital divide.

These concerns were found in higher education, education, language teaching, research and writing, assessment practices, among others. Despite, these concerns largely overlap across different studies, some variations were evident. In higher education, challenges and concerns included plagiarism, digital divide, fairness, responsibility, lack of knowledge and

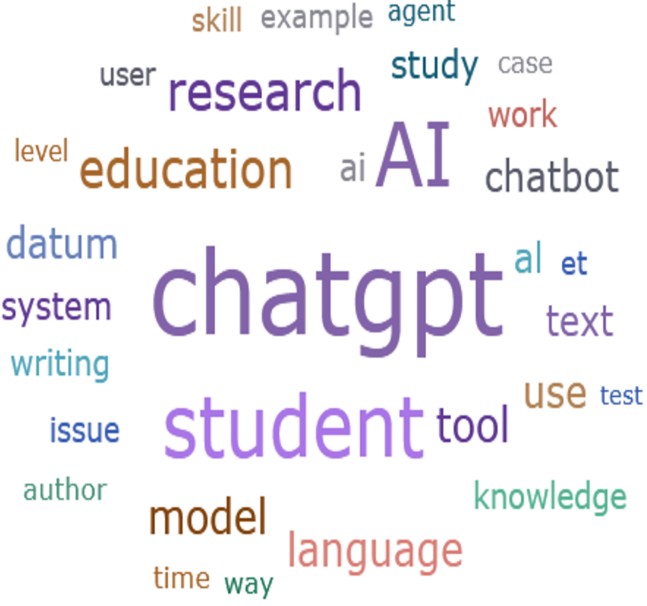

**Figure 1 Concept in the analyzed literature.**

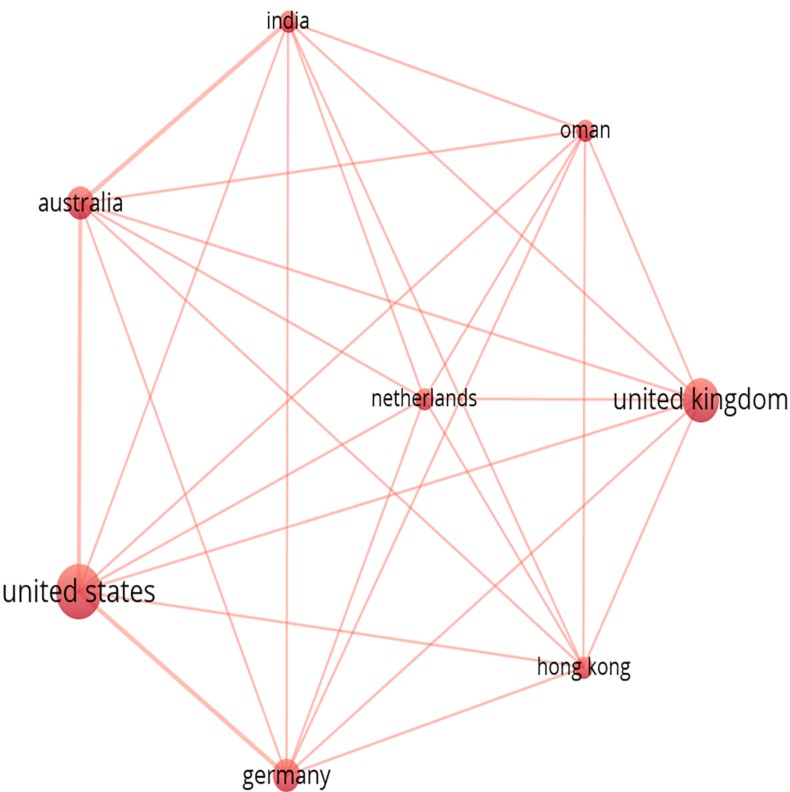

**Figure 2 Top collaborating countries.**

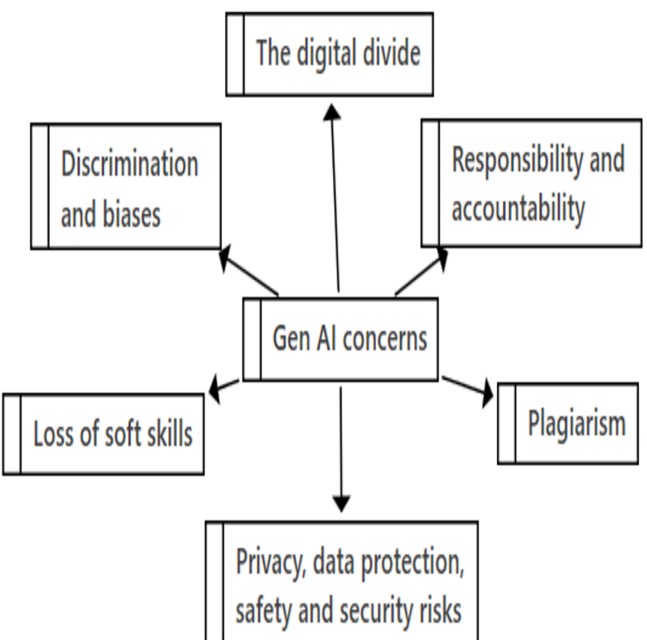

**Figure 3 Main challenges related to Gen AI in education.**

resources, accuracy, privacy, reliability, biases, and the dissemination of falsified information and to name a few. For instance, *Cotton, Cotton & Shipway (2024)* examined the opportunities and challenges of using ChatGPT in higher education, raising concerns regarding plagiarism and academic dishonesty. Similarly, *Tlili et al. (2023)* study revealed worries regarding cheating, honesty, reliability of responses, privacy, dissemination of misleading information, and potential manipulation. While some studies such as *Dalalah & Dalalah (2023)* emphasized on the challenge of false positive and false negative detection in generative AI, with a focus on ChatGPT in education and academic research. They revealed significant issues with false detection.

In the realm of medical education such as health professions education (HPE), concerns were related to data collection, anonymity, privacy, consent, data ownership, security, bias, transparency, responsibility, autonomy, and beneficence (*Masters, 2023*), while in research domain, concerns revolve around fabricated content like citations. For instance, in a study by *McGowan et al. (2023)*, ChatGPT generated thirty-five citations, with only two being genuine. Of the rest, 12 resembled real manuscripts but had errors, while 21 were a mix of existing manuscripts. In addition, non-peer-reviewed journals and online sources can contain inaccurate or outdated information, which Gen AI tools may present as established knowledge (*UNESCO, 2023*). It is further stated that this can pose a risk of producing low-quality research article and diminishing the quality of scientific publications, which could, in turn, affect future AI training data. Novice researchers may particularly encounter obstacles when they lack an understanding of how it generates outputs (*Dwivedi et al., 2023*). In addition, Gen AI like ChatGPT poses a significant challenge for the academic community by questioning output veracity and authorship

authentication, undermining the current publishing hierarchy reliant on journal quality and reputation (*Dwivedi et al., 2023*).

In second language writing, the concerns were related to writing pedagogy, academic integrity, and loss of uniqueness in writing style. For example, *Barrot (2023)* investigated the utilization of ChatGPT for second language writing, noting concerns among academics regarding its potential impact on writing pedagogy and academic integrity, despite acknowledging its potential as an effective tutor and source of language input. Furthermore, assessment practices were also affected, with concerns evident in this area. For instance, *Farazouli et al. (2023)* investigated how AI chatbots, particularly ChatGPT, influence university teachers' assessment practices, focusing on home examinations in undergraduate contexts. Findings revealed varying passing rates for chatbot-generated texts and suspicions regarding their authenticity, with teachers exhibiting patterns of downgrading when grading student-written texts.

Some studies had a broader focus. For example, *Stahl & Eke (2024)* focused on the ethical dimensions surrounding generative conversational AI systems. Key issues highlighted included responsibility, inclusion, social cohesion, autonomy, safety, bias, accountability, and environmental impacts. Next, we discuss the reflections on Gen AI in education and strategies for addressing the identified concerns in detail.

### RQ2. How to address the challenges posed by Gen AI like ChatGPT in education?

To address research question 2, an in-depth discussion of the identified challenges along with reflections and solutions or strategies, is presented below. The authors may or may not have reported one or multiple reflections and/or solutions in the given tables.

### Plagiarism

Plagiarism, the most prevalent concern, poses a significant challenge regarding the utilization of Gen AI in education, as noted in 16 publications (see Table 1). For instance, ChatGPT ability to produce original content that may evade detection by anti-plagiarism checkers, particularly when rephrased using tools like Quilbolt is a concern (*Barrot, 2023*). While ChatGPT can assist students in text creation, it can also lead to fraud when a text is generated by ChatGPT and presented as a learner's own work (*Dwivedi et al., 2023*). They assert that controlling learners' plagiarism and cheating in assignments, theses, and dissertations becomes more challenging due to ChatGPT advanced capabilities, making it difficult for teachers to detect assignments generated by the tool.

In addition, the lack of originality may lead to adverse outcomes. For example, ChatGPT potential to generate persuasive but often inaccurate article could distort scientific facts and encourage plagiarism, challenging traditional review processes and reshaping how research results are verified (*Dwivedi et al., 2023*). Instructors were reported about the potential increase in plagiarism and the added complexity of evaluating work (*Cardon et al., 2023*). Institutions encounter challenges in detecting and penalizing fraud, with current strategies lacking legal validity and internal regulations often failing to address AI-generated content fraud (*Dwivedi et al., 2023*). This creates disparities between

**Table 1 Plagiarism.**

| Concern | Reflections | Solutions | References |
|---|---|---|---|
| Plagiarism | Plagiarism prevalence<br>Academic fraud<br>Lack of originality<br>Impact on review<br>Unrefereed sources<br>Plagiarism detection issue<br>Plagiarism tool limitations<br>Research quality issues<br>Publishing problems<br>Legal and institutional challenges | Domain-specific AI systems<br>Plagiarism detectors<br>AI-powered plagiarism detection<br>Awareness and consequences<br>Declarations of originality<br>Draft submissions<br>Guidelines for Gen AI<br>Student work monitoring<br>Anti-cheating measures<br>enforcement | *Barrot (2023)*, *Bom (2023)*, *Cardon et al. (2023)*, *Chan & Hu (2023)*, *Choudhary & Pandita (2023)*, *Cotton, Cotton & Shipway (2024)*, *Dalalah & Dalalah (2023)*, *Dwivedi et al. (2023)*, *Eke (2023)*, *Farazouli et al. (2023)*, *Farrokhnia et al. (2023)*, *Kasneci et al. (2023)*, *Kooli (2023)*, *McGrath et al. (2023)*, *Stahl & Eke (2024)*, *Tlili et al. (2023)* |

penalized students and undetected ones, democratizing plagiarism (*Farrokhnia et al., 2023*). However, this scenario is expected to evolve as ChatGPT and its counterparts are likely to advance in the months and years ahead (*Dwivedi et al., 2023*). Ways for mitigating plagiarism can include raising awareness in students regarding the consequences of Gen AI (*Cotton, Cotton & Shipway, 2024*), developing domain-specific AI systems (*Farazouli et al., 2023*), advancing existing plagiarism detection software (*Kooli, 2023*), setting explicit Gen AI usage guidelines, requesting pre-submission drafts for review from students, and closely monitoring learners' work (*Cotton, Cotton & Shipway, 2024*), using anti-plagiarism tools (*Barrot, 2023*; *Cotton, Cotton & Shipway, 2024*; *Eke, 2023*; *Kooli, 2023*), and incorporating or supplementing integrated AI detector with current anti-plagiarism software (*Dalalah & Dalalah, 2023*; *Dwivedi et al., 2023*).

## Responsibility and accountability

Table 2 highlights responsibility and accountability concerns raised in 15 publications, which are pertinent to developers, facilitators, and users. However, a significant challenge arises from Gen AI developers often disclaiming responsibility for output such as text, images, audio, video, recommendations, predictions, and other type of data or information, particularly when violating national or international laws on copyright or intellectual property rights. For instance, OpenAI disclaims responsibility for intellectual property infringements stemming from ChatGPT use, presenting researchers with challenges in preventing such infringements (*Dwivedi et al., 2023*). They further state that the issue arises during the generation process when the model produces answers that contain full sentences or paragraphs that it has seen in the training set leading to copyright infringement and plagiarism. The terms of use for AI tools often protect the companies creating these tools from being held responsible for any inaccuracies or unreliability in the results generated by the AI (*UNESCO, 2023*). It is further stated that achieving accountability in such cases is difficult, especially when there are no well-defined governance rules outlining accountability requirements. Legal complexities, like copyright lawsuits such as Stability AI and Getty Images, raise questions about responsibility—whether it lies with the developer, user, or AI itself (*UNESCO, 2023*). In this case, the

**Table 2 Responsibility and accountability.**

| Concern | Reflections | Solutions | References |
|---|---|---|---|
| Responsibility and accountability | Open AI disclaims responsibility<br>Training data quality and copyright infringement<br>Responsibility and accountability challenges<br>Legal complexity of AI-generated content<br>The risk of false information<br>Susceptibility to errors<br>Dissemination of inaccurate information and misinformation<br>Potential harm to users<br>Unnoticed errors in academic articles<br>Inaccurate responses<br>Fabricated content | Protocols for sustainable, ethical, and respectful usage<br>Permission from original document authors<br>Terms of use for model-generated content<br>Policy communication to users<br>Institutions guidelines and policies<br>Transparency and permissions<br>Author responsibility guidelines<br>Accountability<br>Higher AI and information literacy<br>Human verification/validation<br>Empirical data training test<br>Transparency, versatility, and variability<br>Suitable oversight and measures | *Bom (2023)*, *Cardon et al. (2023)*, *Dalalah & Dalalah (2023)*, *Dwivedi et al. (2023)*, *Eke (2023)*, *Farrokhnia et al. (2023)*, *Kasneci et al. (2023)*, *Kooli (2023)*, *Masters (2023)*, *McGowan et al. (2023)*, *McGrath et al. (2023)*, *Rasul et al. (2023)*, *Stahl & Eke (2024)*, *Tlili et al. (2023)*, *UNESCO (2023)* |

UNESCO Recommendations on AI state that AI systems should not be granted legal personality and that AI actors bear ethical responsibility and liability for their roles in AI systems (*UNESCO, 2023*). Therefore, AI developers, users, and policymakers need to collaborate and adapt existing legal frameworks to ensure that ownership, copyright, and other intellectual property concerns are appropriately addressed in the context of AI-generated works. For copyright protection, legal professionals with expertise in intellectual property and AI law are valuable resources for navigating these evolving legal landscapes. Protocols have to be developed to ensure sustainable, ethical, and respectful use of these tools, especially regarding copyright and sources (*Vázquez-Cano et al., 2023*). Furthermore, there is a need for suitable oversight and accountability measures (*UNESCO, 2023*). For content usage, transparency in seeking permission from original document authors during model training, adhering to copyright terms for open-source content, specifying terms of use for content generated by the model, and informing users about these policies are crucial strategies (*Kasneci et al., 2023*). For journals, it is important to establish clear guidelines. For example, these may include checklist and mechanism for authors' and reviewers' responsibility (*Bom, 2023*).

In addition, there are concerns regarding the risk of false information, fabricated content, and false positives and false negatives. The concerns encompass the spread of false information, including fake information, misleading information, misinformation, or disinformation (*Cardon et al., 2023*; *Dwivedi et al., 2023*; *McGowan et al., 2023*), fabricated content (*Dwivedi et al., 2023*; *McGowan et al., 2023*) and false positive and false negative (*Dalalah & Dalalah, 2023*; *Kooli, 2023*). These arise when the model is exposed to false or misleading information during training, causing it to generate inaccurate or unreliable responses (*Dwivedi et al., 2023*) as the quality of Gen AI outputs hinges on the quality of its training data. Gen AI like ChatGPT is susceptible to errors, including the dissemination of inaccurate information (*Tlili et al., 2023*) as it generates content without true comprehension, presenting both accurate and false information with equal confidence (*Dwivedi et al., 2023*). Harmful behaviors that may be exhibited by ChatGPT like

dishonesty, manipulation, and misinformation could potentially harm users, especially those with limited information and communication technology (ICT) knowledge, rather than providing them with assistance (*Tlili et al., 2023*). This becomes highly misleading, when users lack expertise or background knowledge in the field. As professionals increasingly use AI for their communication, they will likely encounter reduced tolerance for inaccuracies, misinformation, and other forms of unreliable communication (*Cardon et al., 2023*). They further maintain that professionals must prioritize advanced information literacy and improved discernment for distinguishing between accurate and inaccurate information always fully upheld. In addition, Gen AI like ChatGPT ability to produce persuasive but often inaccurate content, leading to distortions of scientific facts and the spread of misinformation (*Dwivedi et al., 2023*). For instance, when requested to provide references or citations, they often fabricated sources to support their output (*UNESCO, 2023*). The inaccurate or unreliable responses can have significant consequences, especially when ChatGPT information is relied upon, such as in decision-making and information dissemination contexts (*Dwivedi et al., 2023*). Therefore, designing a responsible chatbot (*Tlili et al., 2023*) that focuses on transparency, explainability, and variability (*UNESCO, 2023*) with empirical data training (*Dwivedi et al., 2023*), is imperative. Additionally, users must utilize Gen AI like ChatGPT with responsibility and accountability, as previously mentioned, by prioritizing information literacy (*Cardon et al., 2023*; *UNESCO, 2023*), ethical considerations (*Dwivedi et al., 2023*), and the necessity for human verification and validation (*Cardon et al., 2023*; *Kasneci et al., 2023*), to mitigate the risk of manipulation, misinformation, and disinformation.

Furthermore, noted are concerns about false negatives and false positives (*Dalalah & Dalalah, 2023*; *Dwivedi et al., 2023*). This can impact academic integrity in plagiarism detection. False negatives refer to undetected plagiarism instances, while false positives involve incorrectly flagging legitimate content as plagiarism. For example, a student's original work with a writing style that resembles that of AI-generated text could be wrongly labeled as plagiarism (false positive), or text that lacks originality may be flagged as original (false negative). In order to mitigate the risk of false information and/or fabricated content, implementing strategies such as raising awareness (*Dalalah & Dalalah, 2023*), advancing information literacy (*Cardon et al., 2023*), conducting human verification (*Cardon et al., 2023*; *Kasneci et al., 2023*), and incorporating ethical reflections and empirical data training tests (*Dwivedi et al., 2023*).

## Privacy, data protection, safety, and security risks

Using large language models in education raises privacy, data protection, safety and/or security concerns as indicated in Table 3 with 13 studies. Gen AI like ChatGPT use of ML algorithms involves processing vast amounts of data, making it susceptible to cyberattacks. The concerns extend to potential data breaches, unauthorized access, and misuse of student data (*Kasneci et al., 2023*). For example, it generates extensive data on students' academic performance, learning preferences, and personal information (*Suen & Hung, 2023*). This sensitive information can be at risk of theft, potentially resulting in security threats (*Dwivedi et al., 2023*), and risk of privacy and data protection (*UNESCO, 2023*).

**Table 3  Privacy, data protection, safety, and security.**

| Concern | Reflections | Solutions | References |
|---|---|---|---|
| Privacy, data protection, safety, and/or security | Vulnerability to cyberattacks<br>Risks of data breaches<br>Unauthorized access<br>Sensitive and personal<br>Information exposure<br>Privacy and security<br>Dangerous content<br>Cyber threats<br>Accuracy of content | Responsible and ethical data usage<br>  Robust data privacy and security policies compliant with<br>  ethics and relevant national and international law<br>  Mandatory consent for transparent data practices<br>  Modern technologies for data protection<br>  Regular security audits<br>  Data breach incident response plan<br>  Staff education and awareness on data privacy and security<br>  Age appropriateness | *Barrot (2023)*, *Cardon et al. (2023)*, *Chan & Hu (2023)*, *Dalalah & Dalalah (2023)*, *Dwivedi et al. (2023)*, *Giannini (2023)*, *Kasneci et al. (2023)*, *Kooli (2023)*, *Masters (2023)*, *Stahl & Eke (2024)*, *Tlili et al. (2023)*, *UNESCO (2023)* |

Researchers reported concerns about personal data exposure during interactions with ChatGPT, potentially revealing sensitive details (*Tlili et al., 2023*). Thus, digital technologies can affect individuals in terms of privacy, autonomy, identity, and security along with broader societal consequences (*Dwivedi et al., 2023*). Furthermore, the use of foundation models like ChatGPT poses unforeseen security concerns, such as the dissemination of dangerous content like instructions for creating harmful devices such as dirty bombs, and the potential facilitation of cyber threats through the tool programming capabilities, enabling the creation of viruses, malware, ransomware, spyware, and phishing campaigns (*UNESCO, 2023*).

A series of measures should be implemented to ensure privacy and security encompassing emphasis placed on upholding national and international legal standards in data collection, use, sharing, storage, and deletion, along with the adoption of robust data protection frameworks and governance mechanisms (*UNESCO, 2023*). Furthermore, implementing robust data privacy and security policies, guidelines, principles, and/or strategies that align with established ethical standards and laws (*Kasneci et al., 2023*; *Tlili et al., 2023*) is crucial. In addition, educators should independently evaluate AI applications and build the capacity to approve them for formal school use, rather than relying on corporate creators (*Giannini, 2023*). Ensuring responsible data usage (*Dwivedi et al., 2023*; *Tlili et al., 2023*) and prioritizing transparency with students and families for data collection, storage, and usage are other important measures. Utilize robust measures including anonymization, secure infrastructures, encryption, and regular audits to safeguard data effectively and proactively identify vulnerabilities (*Kasneci et al., 2023*). It is also essential to ensure age appropriateness for AI usage. Presently, the ChatGPT terms of use mandate that users must be a minimum of 13 years old, with individuals under the age of 18 necessitating the consent of their parent or legal guardian to access the services (*OpenAI, 2023*). Promoting education and awareness among staff, including educators and students, about data privacy and security policies, regulations, ethical considerations, and best practices can be another important solution to data protection (*Kasneci et al., 2023*). Lastly, Gen AI should be subjected to rigorous scrutiny based on these and other relevant criteria.

**Table 4 The digital divide.**

| Concern | Reflections | Solutions | References |
|---|---|---|---|
| Digital divide: Inclusion, equity, access, and fairness | Risk of inequality in AI access<br>Potential concentration of power<br>Language-related inequity<br>Worsening educational disparities | Dialectical pedagogy with Gen AI<br>Individualized dialogue support in ill-defined domains<br>Continues update and expansion,<br>Increased access to information,<br>Support for underserved learners<br>Policies provision for Gen AI usage<br>Resource allocation<br>AI models for the benefit of all<br>Multilingual proficiency and multicultural representation<br>Learners with communication disability<br>Openness and optimism<br>Relevance of pedagogical methods | *Cardon et al. (2023)*, *Cotton, Cotton & Shipway (2024)*, *Dwivedi et al. (2023)*, *Giannini (2023)*, *Kasneci et al. (2023)*, *Kocoń et al. (2023)*, *McGrath et al. (2023)*, *Rasul et al. (2023)*, *Stahl & Eke (2024)*, *Tlili et al. (2023)*, *UNESCO (2023)*, *Yildirim-Erbasli & Bulut (2023)* |

## The digital divide

Integrating Gen AI like ChatGPT into education raises concerns about unequal access, leading to disparities including inclusion, equity, access, and fairness as depicted in Table 4. The aspect of whether it causes a digital divide or helps prevent it is highlighted in 12 studies. While AI and digital technologies offer diverse knowledge systems, but if a few dominant AI models and platforms control our access to knowledge, we risk moving in the opposite direction (*Giannini, 2023*), concentrating wealth and power (*UNESCO, 2023*). In other words, instead of increasing diversity and openness in knowledge systems, it might lead to a concentration of power and control in the hands of a few, limiting the available variety and diversity. For example, despite ChatGPT being a multilingual model (*Choudhary & Pandita, 2023*), the variations in accuracy and nuances in language can cause inequity and unequal access among learners. In this case, some students may not benefit fully from ChatGPT due to language limitations or reduced accuracy, creating disparities in their educational experience. Researchers noted language differences in queries, highlighting that even with accurate translation, responses often reflect U.S.-centric perspectives (*Rettberg, 2022*). While efforts to address multilingual fairness are underway, there remains ample room for improvement in this respect (*Kasneci et al., 2023*). When employing AI tools in education, UNESCO insists on prioritizing inclusion, equity, quality, and safety (*Giannini, 2023*). Blocking access to Gen AI like ChatGPT can also result in a digital divide and is not advisable.

However, some publications reported that ChatGPT holds significant potential for providing anytime and anywhere, individualized support, and educational opportunities. *Giannini (2023)* states that we must embrace openness and optimism toward AI potential to enhance, supplement, and enrich formal education. In language learning, for instance, less prepared students, through no fault of their own, can benefit from ChatGPT to enhance their writing skills (*Cardon et al., 2023*). The authors emphasize the utilization of Gen AI, particularly ChatGPT, in facilitating personalized learning, tailored instruction, and learning pathways (*Cardon et al., 2023*; *Cotton, Cotton & Shipway, 2024*; *Dwivedi et al., 2023*; *Farrokhnia et al., 2023*; *Tlili et al., 2023*; *Vázquez-Cano et al., 2023*;

**Table 5 Discrimination and biases.**

| Theme | Reflections | Solutions | References |
|---|---|---|---|
| Discrimination and biases | Biased perpetuations/reflections<br>Impact on students' well-being<br>Impact on educational content<br>Algorithm bias<br>Potential of exploiting<br>Vulnerable populations<br>Lack of representation | Representative and diverse training data<br>Continuous bias evaluation<br>Fairness measures and bias correction<br>Transparent mechanisms<br>Ongoing model updates<br>Guarding against data and algorithm bias<br>Bias detection and mitigation method | *Cardon et al. (2023), Choudhary & Pandita (2023), Dalalah & Dalalah (2023), Dwivedi et al. (2023), Farrokhnia et al. (2023), Giannini (2023), Kasneci et al. (2023), Kooli (2023), Masters (2023), Stahl & Eke (2024), Tlili et al. (2023), UNESCO (2023)* |

*Yildirim-Erbasli & Bulut, 2023*). For instance, ChatGPT usage personalized to benefit students with communication disabilities (*Chaudhry et al., 2023*) by comprehending poorly written text, contributing to their language improvement. In addition, opportunities include accessibility of large language models like ChatGPT (*Dwivedi et al., 2023*), multilingual support (*Choudhary & Pandita, 2023*), and text-to-speech and speech-to-text capabilities (*Kasneci et al., 2023*). These opportunities and/or advantages extend to other areas such as writing support (*Dwivedi et al., 2023*), increased information access (*Farrokhnia et al., 2023*), and aiding individuals without in-person tutors (*Su & Yang, 2023*).

However, policies and regulations should be in place to ensure equal access to GenAI, promote multilingualism, facilitate boundary-free access, and address the needs of learners with economic problems, disabilities and others. For instance, policies may include subsidies, public access programs, government procurement incentives, grants, and other initiatives aimed at fostering inclusion, and equity in AI access (*Stahl & Eke, 2024*). Furthermore, the promise of AI for all (*Kasneci et al., 2023*) and continuous updates and expansion (*Barrot, 2023*) are central to promoting universal access. Equitable education on a broad basis (*McGrath et al., 2023*) and equity-oriented means for educational entities (*Kasneci et al., 2023*) further contribute to learning for all. In addition, it is imperative that educational chatbots adhere to user-centered design principles. Finally, governmental regulation of AI training, as advocated by *Kasneci et al. (2023)*, can reinforce its equitable inclusiveness, collectively creating a more accessible and fair AI education.

## Discrimination and biases

Discrimination and bias was another most notable concern, highlighted in 12 publications, see Table 5. One of the concern is that the Gen AI models like ChatGPT may perpetuate existing biases, stereotypes, and discrimination in society as it is trained on large corpora of textual data freely available on the internet (*Dwivedi et al., 2023*). This risk arises from biases inherent in the training data, which may manifest in the model outputs (*Dwivedi et al., 2023; Kasneci et al., 2023; Kooli, 2023*). There have been extensive examples of racist, sexist, ableist, and other discriminatory language making its way into the model and is then generated as output (*Dwivedi et al., 2023*). Gender bias, particularly, has been highlighted in prior research, with GPT-3 narratives reinforcing stereotypes about females, depicting them as less powerful and defining them by physical appearance and family roles

**Table 6 Loss of soft skills.**

| Concern | Reflections | Solutions | References |
|---|---|---|---|
| Loss of Soft skills: Communication, problem-solving, creativity, decision-making, critical thinking, reasoning, and/or research skills | Reduced motivation for independent work Risk to cognitive abilities Reduced opportunities for human learning Impact on (social) interactions | Monitoring language model usage Efficient human resource allocation Instructor role Rethinking instruction design and assessment Digital technology-supported cognitive skills activities Performance-based evaluation system Creative and independent projects Creative use of large language models New ways of teaching Varying information resources Human oversight AI literacy/awareness Investing in and promoting digital, media, and information literacy skills | *Barrot (2023)*, *Cardon et al. (2023)*, *Chan & Hu (2023)*, *Choudhary & Pandita (2023)*, *Dwivedi et al. (2023)*, *Farrokhnia et al. (2023)*, *Kasneci et al. (2023)*, *Kooli (2023)*, *Stahl & Eke (2024)*, *UNESCO (2023)*, *Yildirim-Erbasli & Bulut (2023)* |

(*Ary et al., 2018*). These inadvertent biases and discriminatory elements can profoundly impact students' well-being, leading to stress and anxiety, diminished academic performance due to discouragement, erosion of self-esteem and self-confidence, and a weakened sense of belonging in the learning environment. In addition, the biased trained data might reflect societal value judgments by promoting certain beliefs or opinions over others, potentially impacting the objectivity of educational content. Similarly, if the training data lacks representation from underrepresented groups, the tools might present biased information about their experiences or contributions. In these cases, the educational quality and equity might be compromised.

Stakeholders need to be aware of these biases and work towards ensuring that AI tools like ChatGPT provide unbiased information to support students' learning and promote a well-rounded education. Thus, for the model to provide accurate and unbiased responses, it is imperative to use a high-quality dataset that is representative of the question being posed to it because any biases or inaccuracies in the data can be reflected in the model output (*Dwivedi et al., 2023*). Other vital strategies include ensuring model fairness and mitigating biases through transparency (*Kasneci et al., 2023*; *Masters, 2023*), training with diverse data, ongoing performance monitoring, implementing fairness measures, human oversight, providing educator training, and conducting regular expert-supervised updates (*Kasneci et al., 2023*).

## Loss of soft skills

Table 6 highlights concerns about the excessive reliance on Gen AI like ChatGPT, potentially impeding the development of soft skills such as critical thinking, creativity, problem-solving, decision-making, reasoning, and/or meaningful face-to-face interactions across 11 studies. For example, relying too much on ChatGPT may undermine higher cognitive abilities such as creativity and problem-solving, diminishing motivation for independent work (*Farrokhnia et al., 2023*). This concern is heightened when students lack guidance on appropriate ChatGPT usage (*Barrot, 2023*) or engage in what *Dwivedi et al.*

*(2023)* referred to as "blind usage." While initial utilization of Gen AI models may enhance individual performance, prolonged reliance on AI has been associated with lower performance attributed to the loss of complementary skills (*Cardon et al., 2023*). It can potentially undermine various cognitive abilities linked to literacy, including writing, understanding, and critical thinking (*UNESCO, 2023*), critical thinking (*Tlili et al., 2023*) and creativity (*Kooli, 2023*; *UNESCO, 2023*) and problem solving abilities (*Kooli, 2023*). Moreover, its excessive use can result in reduced opportunities for human learning (*Dwivedi et al., 2023*), and degradation of social interactions (*UNESCO, 2023*), such as teacher-student relationship (*Farrokhnia et al., 2023*).

Similar to that of digital divide, while some studies highlight concerns about excessive dependence on ChatGPT potentially hindering the development of cognitive skills, others reported positive effects. Therefore, we should maintain an open and positive outlook on how AI can enhance and complement the essential learning that occurs through interactions in the physical and social environments of formal education (*Giannini, 2023*). In other words, we should be hopeful and receptive to the idea that AI can contribute positively to education, provided that it is used safely and responsibly. For example, Gen AI, including ChatGPT, has been reported to stimulate critical thinking in students by presenting tailored sets of questions corresponding to their proficiency levels (*Cotton, Cotton & Shipway, 2024*), decision making, and problem-solving and communication (*Dalalah & Dalalah, 2023*). Through the development of customized interactive learning materials, Gen AI such as ChatGPT can foster critical thinking and problem-solving abilities (*Dwivedi et al., 2023*). For example, instructors can leverage Gen AI to create personalized assignments that require critical thinking, problem-solving, and communication skills, such as group discussions, presentations, and interactive activities (*Cotton, Cotton & Shipway, 2024*). Similarly, students could be encouraged to evaluate Gen AI responses and compare them to original content, thereby fostering critical thinking (*Dwivedi et al., 2023*; *Su & Yang, 2023*) as they need to think critically to ensure the accuracy of their writing (*Su & Yang, 2023*).

In the context of written communication and language improvement, students can leverage Gen AI tools like ChatGPT to refine initial outlines, content, organization, and structural feedback, thereby enhancing language style, vocabulary, and grammar (*Barrot, 2023*). In communication, the role of Gen AI can encompass various functions, serving as a tool, assistant, monitor, coach, or teammate (*Cardon et al., 2023*). However, the concerns are that learners will be spoon-fed, and communication will be depersonalized (*Cardon et al., 2023*). In addition, there can be resistance to change in teaching (*Cardon et al., 2023*). Moreover, as AI advances in language abilities, it challenges our conceptions of what it means to be human and how we coexist with increasingly intelligent machines (*Giannini, 2023*). However, at the same time, this indicates an inconsistency, underscoring the need for further empirical research to better understand the overall impact of Gen AI on cognitive skills and to identify effective strategies for its integration into education.

It is worth highlighting that while some Gen AI like ChatGPT does not demand extensive technical ICT proficiency, it heavily relies on critical thinking and question-asking skills for optimal performance (*Tlili et al., 2023*). Without appropriate prompts, the

results it generates can be misleading. In addition, it has ushered in fresh opportunities for interdisciplinary collaboration in academic research. Through its automation of language-related tasks, researchers from diverse fields can now collaborate with greater efficiency, potentially resulting in the emergence of novel research concepts, and methodologies, and a surge in innovation and breakthroughs within academic research (*Dwivedi et al., 2023*).

The strategies involve future professionals cultivating AI literacy in authenticity, accountability, and agency, with a focus on information literacy and communication reliability (*Cardon et al., 2023*). They further state that instructors must consistently assess the effectiveness of Gen AI for diverse professional objectives, continually enhance their expertise in every aspect of AI literacy, and refrain from isolated Gen AI endeavors. Other recommendations that can be implemented include raising awareness of AI impact (*Dwivedi et al., 2023*), particularly about the limitations and vulnerabilities. Encouraging students to assess and provide feedback on ChatGPT responses to questions can also contribute to critical thinking (*Dwivedi et al., 2023*), and diversifying information sources, such as books and articles, can contribute to critical thinking. Furthermore, integrating critical thinking and problem-solving activities into the curriculum can help students develop these crucial skills. Educators should actively promote critical thinking when students engage with technology (*Dwivedi et al., 2023*). Lastly, human expertise and teacher involvement should play a central role in reviewing, validating, and explaining AI-generated information to ensure accuracy, authenticity, and responsible use. These recommendations collectively foster a balanced and responsible usage of Gen AI in education while nurturing critical thinking and ethical practices.

## Critical evaluation of the solutions: theories and authors' perspective

In this section, we critically evaluated the synthesized strategies to the challenges posed by Gen AI (ChatGPT) in education. The synthesis (see, Fig. 4.) encompasses various concerns, including plagiarism followed by responsibility, accountability. Other concerns included the risk of false information, the digital divide; and privacy, data protection, safety, and security, discrimination and bias, and loss of soft skills. The extracted solutions were then assessed as displayed in Table 7. The table consists of columns for challenges, solutions, practicality rating, theoretical validation, and authors' perspective. The practicality rating indicates the feasibility or applicability of derived solutions in educational settings. The theoretical validation column shows assessment of the solutions against the established theories from the authors' perspective. Three experts, two from the Center for Lifelong Learning and one from the School of Digital Science, assessed and validated the solutions. The category concerning plagiarism is evaluated through deontology, a theory associated with Kant, which prioritizes moral rules and duty over consequences (*Anshari et al., 2023*). Deontology holds the normative ethical notion that actions are inherently right or wrong, significantly influencing decision-making approaches (*Sheedy, 2024*). It focuses on duty and defines the morality of actions based on strict principles, categorizing them as obligatory, prohibited, or permissible (*Banks, 2015*; *Sandberg, 2013*). In deontological ideology, judgment relies on implicit obligations, rules, and responsibilities (*Cohen, Pant & Sharp, 2001*; *Turk & Avcilar, 2018*). A student who

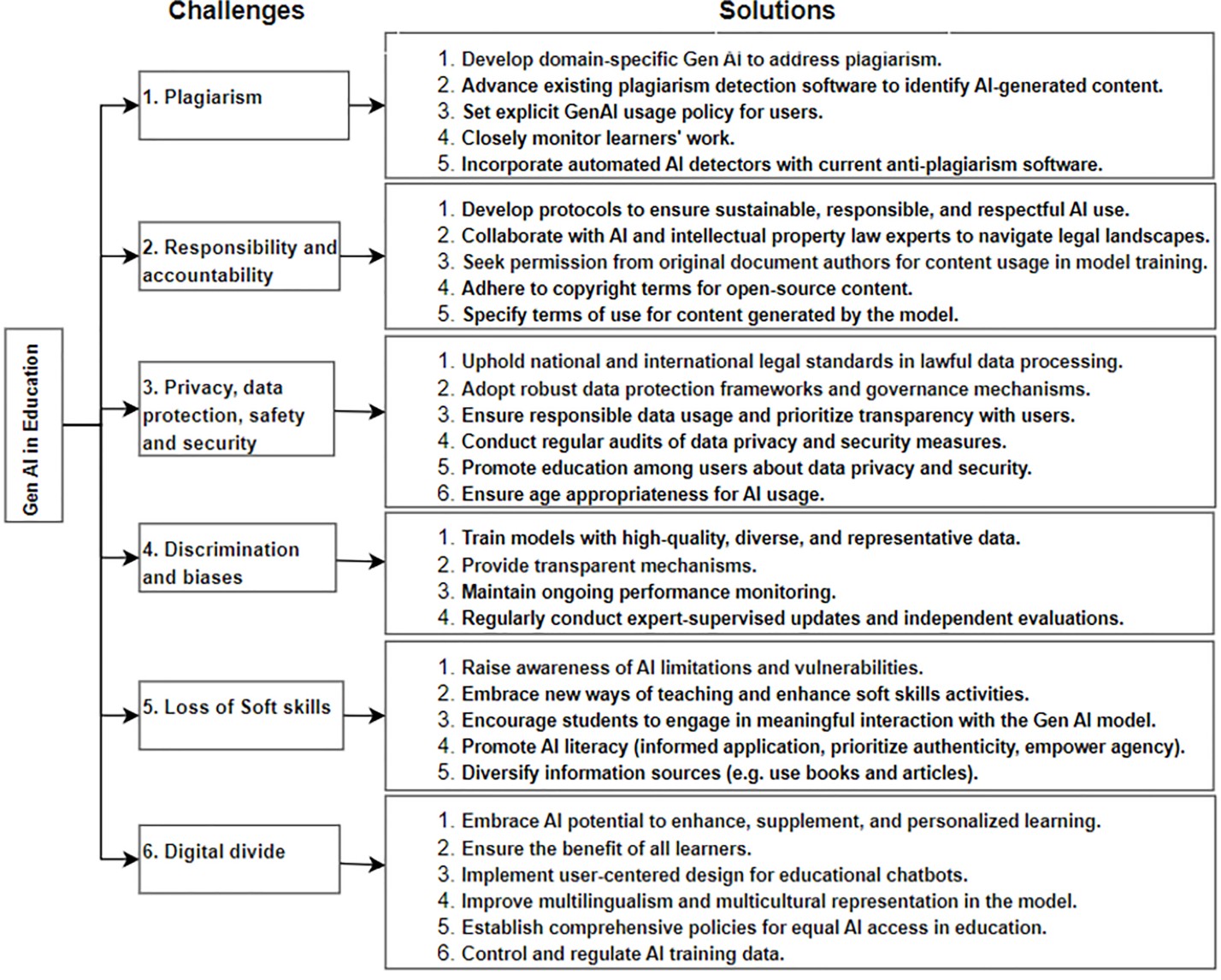

**Figure 4 Synthesis of solutions for addressing emerged challenges.**

adheres to deontology could perform better academically by fulfilling his or her essential responsibility as a student (*Jung, 2009*). The integration of justice and deontology ideologies enhances moral meaningfulness, which, in turn, fosters student citizenship behaviors and in-role performance, ultimately positively influencing academic achievement (*Soto-Pérez, Ávila-Palet & Núñez-Ríos, 2022*). Ethical issues in AI are analyzed within this framework to provide insights for policy recommendations (*Anshari et al., 2023*).

In the category of plagiarism, solutions 3 was rated as highly practical, while solutions 1, 2, 4, and 5 were deemed practical. The strategies, developing domain-specific Gen AI and advancing existing plagiarism detection software to address plagiarism in various fields

**Table 7 Synthesis and critical evaluation.**

| No. | Challenges | Solutions | Practicality rating (H, P & M) | Theoretical validation | Authors' perspective |
|---|---|---|---|---|---|
| 1. | Plagiarism | 1 & 2 | P | Deontology | Supported |
| | | 3 | H | Deontology | Supported |
| | | 4 & 5 | P | Deontology | Supported |
| 2. | Responsibility and accountability | 1, 3–5 | H | Deontology | Supported |
| | | 2 | P | Deontology | Supported |
| 3. | Privacy, data protection, safety and security | 1–6 | H | GDPR | Supported |
| 4. | Discrimination and biases | 1 | P | Intersectionality theory | Supported depends on the availability of diverse and quality data and its effective use |
| | | 2 | H | Intersectionality theory | Supported |
| | | 3 & 4 | P | Intersectionality theory | Supported |
| 5. | Loss of soft skills | 1 | H | Deontology | Supported |
| | | 2 | P | Social constructivism | Supported with the need for certain modifications to the curriculum |
| | | 3 | H | Social constructivism | Supported with the requirement for an investment in technology infrastructure |
| | | 4 & 5 | H | Connectivism | Support |
| 6. | Digital divide | 1 & 6 | H | UDL | Support |
| | | 2 | P | UDL | Supported |
| | | 3 | P | UDL | Supported though careful consideration and adaptation for diverse user needs are required |
| | | 4 | M | UDL | Supported with appropriate efforts and right resources |
| | | 5 | M | UDL | Supported depends on government and organizational capabilities and coordination levels |

**Note:**
H, Highly Practical; P, Practical; M, Moderately Practical.

reflects the duty to maintain truthfulness and respect for intellectual property. Setting explicit guidelines for Gen AI usage is another strategy for the responsible use of AI, which underscores the importance of providing ethical guidance and ensuring adherence to academic integrity principles. The strategy, closely monitoring learners work align with the duty to verify the originality of work and promote honesty among learners. While monitoring is practical for ensuring academic integrity, it must be conducted with respect for privacy and individual rights, balancing the duty of monitoring with ethical considerations. Incorporating automated AI detectors with current anti-plagiarism software, reinforces the commitment to uphold ethical standards within educational contexts. These strategies prioritize the duty to support academic honesty and ethical conduct while addressing the issue of plagiarism.

Regarding responsibility and accountability, solutions 1, 3, 4, and 5 were considered highly practical, and solution 2 was rated practical. The strategy, developing protocols to

ensure sustainable, responsible, and respectful use of AI tools underlines the obligation to act ethically and responsibly. Collaborating with AI and intellectual property (IP) law experts to navigate legal landscapes demonstrates a commitment to following the rules and laws, aligning with deontological principles. Additionally, seeking permission from original document authors for content usage and adhering to copyright terms for open-source content strategies reflect the ethical duty to respect the rights of content creators. Specifying terms of use for AI-generated content and informing users about these terms illustrate a commitment to transparency and accountability, which are essential elements of ethical responsibility. These strategies uphold deontological principles by prioritizing honesty, respect, and ethical duty in the context of Gen AI tools development and usage.

The category pertains to privacy, data protection, safety, and security, with the solutions evaluated in alignment with the European Union General Data Protection Regulation (GDPR) (*Bowen, 2021*; *Gautam & Jahankhani, 2021*). All strategies (1–6) were rated as highly practical. The solution emphasizes the importance of upholding national and international legal standards in data collection, use, sharing, and storage, ensuring compliance with GDPR requirements for lawful data processing (*Gautam & Jahankhani, 2021*). The adoption of robust data protection frameworks and governance mechanisms that align with ethical standards and laws reflects the GDPR emphasis on the need for safeguards in the processing of data. Moreover, ensuring responsible data usage and prioritizing transparency with users align with GDPR principles, highlighting the importance of transparency in data processing, as stated by *Schneider & Xhafa (2022)*. Furthermore, conducting regular audits of data privacy and security measures is essential, along with promoting education among users about data privacy and security to guide the utilization of Gen AI reflects GDPR emphasis on ensuring the security and integrity of personal data and transparency and accountability. Ensuring age appropriateness for AI usage can be an effective strategy. Implementing age-appropriate restrictions is essential for AI developers to uphold legal compliance, prevent access to their personal data, promote responsible use of AI systems, and build trust with users and stakeholders.

The category of discrimination and biases is evaluated with Intersectionality theory. This theory, introduced by Kimberlé Crenshaw, acknowledges individuals' multiple intersecting social identities (*e.g.*, race, gender, class, disability, religion, age) that influence their experiences (*Chowdhury & Okazaki, 2020*). Solution 2 is rated as highly practical, while the remaining solutions 1, 3, and 4 are considered practical. To ensure that Gen AI systems produce accurate and unbiased output, training the models with high-quality, diverse, and representative data is an essential solution. This strategy aligns with Intersectionality theory, recognizing that individuals with intersecting identities may have unique experiences that should be reflected in the data. This solution addresses biases associated with social identities, making it less likely for Gen AI systems trained on such datasets to perpetuate biases. Additionally, providing transparent mechanisms are crucial for ensuring openness and clarity in how Gen AI makes decisions, operates, and generates outputs. This makes the processes and decision-making mechanisms understandable and interpretable to humans. Ongoing performance monitoring is also necessary to ensure that Gen AI remains aligned with its intended objectives and helps to detect and correct any

emerging biases. Furthermore, regular expert-supervised updates are crucial to keeping the Gen AI up-to-date. These updates ensure its continued relevance and accuracy, enabling it to adapt to changing circumstances and evolving user needs effectively.

In the loss of soft skills category, the solutions were evaluated considering Deontology, Social Constructivism and Connectivism theories, with solutions 1, 3–5 rated as highly practical while solution 2 received a practical rating. Social constructivism, a subset of constructivism, emphasizes learning through social interaction, distinguishing itself by placing greater emphasis on this aspect (*Kukla, 2000*). This theory asserts that learners actively construct knowledge through interactions with others (*Rannikmäe, Holbrook & Soobard, 2020*; *Vygotsky, 1962*). In addition, Connectivism was employed, a learning theory emphasizing the significance of diverse information sources and networking in the learning process. The theory is built upon four foundational principles: autonomy (learner's choice and control), connectedness (learning through connections with others and resources), diversity (exposure to a wide range of perspectives and ideas), and openness (access to information and knowledge sharing) (*Corbett & Spinello, 2020*).

Awareness regarding the risks of overreliance on Gen AI among users can be raised by experts, educators, and teachers through various communication channels, including digital platforms like websites and blogs, social media, podcasts, webinars, as well as face-to-face methods like workshops and seminars. This aligns with deontological principles emphasizing moral duties and obligations. The strategy, embracing new teaching methods and enhancing soft skills activities (*e.g.*, assigning analytical tasks), aligns with the constructivist principles of prioritizing a student-centered approach. While it may require adjustments to the curriculum, enhancing soft skills activities such as assigning analytical tasks, presenting complex decisions or dilemmas, and creating conflict scenarios is relatively straightforward to apply. Encouraging students to foster meaningful interaction with AI models, such as active engagement, asking insightful questions, receiving instant feedback, critically evaluating its responses, and to name a few can be another promising strategy. It adheres to the principles of social constructivism, which stress the importance of social interaction and the external environment, especially in the context of Gen AI, for knowledge formation (*Vîșcu & Watkins, 2021*). It is highly valuable, as it enables users to interact and receive instant feedback around the clock, eliminating the constraints of traditional methods that require adherence to formal schedules, or contend with different time zones. Another reason is Gen AI can efficiently provide feedback and engage a large number of learners simultaneously, making education more accessible and cost-effective. For instance, consider a scenario where learners in a webinar pose questions to a teacher. In this traditional setup, the teacher can only address one question at a time, even if receiving multiple inquiries from participants. In contrast, Gen AI can handle a multitude of questions from numerous learners and respond to them at the same time. However, it depends on the availability of AI tools and platforms suitable for educational use and may require investment in technology infrastructure. Another strategy involves promoting AI literacy, emphasizing informed application, authenticity, accountability, and individual agency, particularly in writing. This is essential to educate users about how to use AI effectively and efficiently. We feel that Gen AI literacy plays a pivotal role in harnessing the

potential of Gen AI, particularly, to elevate learners' writing while preserving their unique writing styles and creativeness. Apart from this, Gen AI can help save valuable time by automating certain facets of the writing process, like grammar and spelling checks, thereby enabling learners and writers to concentrate on more creative and critical aspects. Diversify information sources (*e.g.*, use books and articles) can be another useful strategy. Connectivism theory emphasizes diverse information sources and networking reinforces this strategy. By leveraging a range of sources and connections, learners can access a wealth of perspectives and stay updated with the latest information, enriching their understanding in a rapidly changing world where knowledge can quickly become outdated.

Activities involves assigning projects, exploring content in different formats, engaging in group discussions, and encouraging learners to present topics of interest.

The category of challenges pertains to the digital divide, encompassing exclusion, inequity, unequal access, and unfairness. To address these challenges, six strategies have been derived, evaluated in light of the Universal Design for Learning (UDL), with strategies 1 and 6 rated as highly practical, 2 and 3 as practical, and 4 and 5 as moderately practical. The UDL model is based on three main principles: providing multiple means of representation, action and expression, and engagement (*Rao, 2023*). Essentially, UDL encourages educators to design accessible curricula and learning environments that minimize barriers to learning from the outset (*Griful-Freixenet et al., 2020*). The strategies such as embracing Gen AI potential to enhance, supplement, and personalize learning resonates with UDL core principles of providing multiple means of representation and engagement. Customizing educational content with Gen AI accommodates diverse learners by enabling individualized, adaptive learning experiences. For instance, it generates interactive visual explanations for visual learners and detailed written instructions for those who prefer text-based guidance. Ensuring the benefit of all learners is imperative for promoting inclusivity, equity, and equal access to education, transcending barriers related to race, gender, socioeconomic status, or other characteristics. It is crucial for creating a more inclusive and effective learning environment. Additionally, the strategy which emphasis on user-centered design of Gen AI tools considering (social, emotional, cognitive, and pedagogical aspects), supports the UDL goal of fostering inclusive education. User-centered design can better cater to students with varying abilities, backgrounds, and learning preferences, making education more accessible and effective for all. Such enhancements can significantly improve the learning experience, rendering it more engaging, satisfying, and effective for learners, consequently leading to improved learning outcomes. For instance, chatbots equipped with natural language processing and sentiment analysis can discern when a student is struggling or frustrated and provide emotional support, encouragement, or resources tailored to the student's emotional state.

Furthermore, the strategy that focuses on improving multilingual and multicultural representation in models echoes UDL emphasis on providing multiple means of expression to meet the diverse needs of students. This promotes an equitable learning environment where all students can access content in their preferred language and see their cultural heritage acknowledged. It is essential to acknowledge that students may have diverse linguistic preferences and cultural backgrounds. An example of this is when

learners see their own language, culture, and experiences reflected in the Gen AI content, they are more likely to connect with the material, leading to increased motivation, involvement, and learning.

The strategy establishing comprehensive policies and regulations can ensure equitable access to and responsible usage of Gen AI. The implementation of inclusive initiatives such as subsidies, public access programs, and government incentives can facilitate a multitude of representation methods. These initiatives aim to accommodate diverse learner needs, ensure equitable access to resources, and promote a more inclusive and supportive learning environment. For example, offering financial support for assistive technologies such as screen readers, speech-to-text software, and specialized devices for students with disabilities, coupled with the provision of affordable high-speed internet connections, empowers learners to access and engage with educational content more effectively. Moreover, controlling and regulating AI training data can help in achieving inclusiveness. While control and regulation are important, the practicality can vary depending on the governments' and other organizations' capabilities and the level of coordination with Gen AI developers. Overall, these strategies align with the Universal Design for Learning (UDL) framework, which aims to promote inclusivity and accessibility in education for all learners. In summary, while most of the strategies are consistent with existing theories, some adjustments may be necessary, and their applicability depends on specific contextual factors.

## IMPLICATION, RECOMMENDATIONS, AND FUTURE RESEARCH

Gen AI like ChatGPT is foreseen to have a lasting presence. While there are concerns and challenges associated with Gen AI usage in education, outright banning may not be the most effective strategy. Instead, a more balanced approach involving policies, regulations and guidance for its optimization should be considered, as it offers educational value by enabling personalized and interactive learning. Similarly, *Howell & Potgieter (2023)* state that educators should embrace them as indispensable tools in teaching and learning, akin to modern workplace tools.

The integration of Gen AI in education has the potential to significantly impact cognitive skills such as communication, problem-solving, creativity, decision-making, critical thinking, reasoning, and research skills when appropriately employed. Educators and institutions should carefully consider how to leverage ChatGPT to enhance these skills in students, enriching students' learning experiences, and offering guidance and essential resources. Moreover, there is a need for further empirical research to examine the effects of Gen AI usage, such as ChatGPT, on cognitive skills, as inconsistencies in reported findings were evident. Moreover, prioritizing inclusion, equity, and access is essential, providing equal opportunities for all students, including equitable access to large language models, especially for disadvantaged and marginalized students, and implementing mitigation strategies to ensure justice. Addressing discrimination and biases in ChatGPT use in education is vital to ensure fair learning opportunities for students. Privacy and security measures must be strengthened to safeguard sensitive student information and maintain

the integrity of educational environments. Additionally, educators and institutions should implement effective strategies to detect and prevent plagiarism and copyright infringement when using ChatGPT in educational settings, promoting academic integrity and respecting intellectual property rights.

Future research in the integration of Gen AI in education should prioritize conducting longitudinal studies to assess its long-term impact on students' learning. Additionally, there is a need for the development and implementation of ethical frameworks and guidelines specific to ChatGPT in educational settings, addressing concerns related to discrimination, privacy, plagiarism, and copyright infringement. Research should also focus on how to achieve accessibility and inclusivity for all students, particularly those with disabilities, and explore the role of teacher training programs in effectively leveraging AI tools. it is recommended that future research focus on empirical studies to test the strategies. Conducting empirical studies will validate and refine the solutions, providing valuable insights into their practical applicability. Understanding the implications of ChatGPT on student engagement and satisfaction will be crucial for its successful integration into education.

## LIMITATIONS

This study has several limitations. Firstly, its reliance on a relatively small number of publications may restrict the breadth of perspectives considered. It is possible that our search strategy may have inadvertently excluded relevant literature. Additionally, our focus on literature published in English may have limited the diversity of perspectives represented in the findings and the reliance on sources like the UNESCO website may introduce biases. The focus solely on challenges associated with ChatGPT in education neglects broader applications of Gen AI that could offer valuable insights. Deductive analysis with specific themes from literature may have caused overlooking other important challenges and relevant solutions. In addition, the study focused mainly on challenges without thoroughly considering the potential benefits and positive impacts of integrating Gen AI into educational settings, indicating a need for further exploration in this regard. Hence, it is crucial to interpret the findings carefully. Future research is recommended to address these gaps and provide a more comprehensive understanding of the topic.

## CONCLUSION

The integration of Gen AI into education has undoubtedly raised a number of concerns and challenges including plagiarism as the most prevalent concern, closely followed by responsibility and accountability. Other major concerns included discrimination and bias, and the loss of soft skills, the digital divide and privacy, data protection, safety, and security risks. Various strategies have been put forward to tackle these concerns, and each set of strategies has undergone rigorous evaluation using diverse theories and frameworks, including deontology, social constructivism, connectivism, intersectionality theory, UDL, and GDPR. Most of these strategies are practical and align with the ethical and pedagogical theories, addressing the challenges. However, the success of these strategies rely on their conscientious implementation.

### Funding
This work was supported by Universiti Brunei Darussalam. The funders had no role in study design, data collection and analysis, decision to publish, or preparation of the manuscript.

### Grant Disclosures
The following grant information was disclosed by the authors:
Universiti Brunei Darussalam.

### Competing Interests
The authors declare that they have no competing interests.

### Author Contributions
- Wali Khan Monib conceived and designed the experiments, performed the experiments, analyzed the data, performed the computation work, prepared figures and/or tables, authored or reviewed drafts of the article, and approved the final draft.
- Atika Qazi conceived and designed the experiments, performed the experiments, analyzed the data, performed the computation work, prepared figures and/or tables, authored or reviewed drafts of the article, and approved the final draft.
- Rosyzie Anna Apong performed the computation work, prepared figures and/or tables, authored or reviewed drafts of the article, and approved the final draft.
- Mohammad Tazli Azizan performed the computation work, prepared figures and/or tables, authored or reviewed drafts of the article, and approved the final draft.
- Liyanage De Silva performed the computation work, prepared figures and/or tables, authored or reviewed drafts of the article, and approved the final draft.
- Hayati Yassin performed the computation work, prepared figures and/or tables, authored or reviewed drafts of the article, and approved the final draft.

### Data Availability
  This is a literature review and no new data were created or analyzed.

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
