# Peer review of "Generative AI and future education: a review, theoretical validation, and authors’ perspective on challenges and solutions"

_PeerJ Computer Science, doi:10.7717/peerj-cs.2105_

## Round 0.1 · original submission · Major Revisions

Based on all observations sent by reviewers the paper can be well improved.

**Language Note:** PeerJ staff have identified that the English language needs to be improved. When you prepare your next revision, please either (i) have a colleague who is proficient in English and familiar with the subject matter review your manuscript, or (ii) contact a professional editing service to review your manuscript. PeerJ can provide language editing services - you can contact us at [email protected] for pricing (be sure to provide your manuscript number and title). – PeerJ Staff

Reviewer 1 ·

Basic reporting

The idea present in this paper titled “Generative AI and future education: A review, theoretical validation and authors perspective on challenges and solutions.” However, the authors are suggested to address the following comments while revising the paper.

The authors suggested adding a section on existing Generative AI technologies to provide a quick overview and later provide a rationale for why specifically focusing on “ChatGpt” or it will be better to look at all the Gen AI models in education as the number of studies are very limited.

The authors are suggested to identify and specify the focus of exiting studies from the perspective of domains such as business, computer science, education, arts and others. It is ambiguous if these areas are not mentioned as the challenges may vary based on the domain and also the severity of challenges may also vary.

Suggested to remove the references that are not related to the theme of the paper such as: Qazi, A., Hussain, F., Rahim, N. A., Hardaker, G., Alghazzawi, D., Shaban, K., & Haruna, K. (2019). Towards sustainable energy: a systematic review of renewable energy sources, technologies, and public opinions. IEEE access, 7, 63837-63851.
Abayomi-Alli, O. O., Damaaeviius, R., Qazi, A., Adedoyin-Olowe, M., & Misra, S. (2022). Data augmentation and deep learning methods in sound classification: A systematic review. Electronics, 11(22), 3795.

The quality of all figures needs to be improved.

Carefully review the paper for typos such as line 712 (missing .)

Deontology is first used at line 558, later on it is defined at 682. It needs to be defined first. Similarly for other theories.

Experimental design

The details on #Codes and % are missing. The authors are suggested to add more details to the codes, what code refer to and how they are assigned. Moreover, what is its (%) refers too. What are these codes and % telling us. A detailed argument needed to be built on these percentages form line 215 to 217. It is merely reported as number only. Moreover, how these code frequency occurrences are generated such as 128. What is 128 referring to.
All these details are presently missing in the paper and adding this will give a better understanding to the authors.

Validity of the findings

What is practicality rating in Table 7. It is defined nowhere in the manuscript.

How theoretical validation is performed? Who performed theoretical validations and what are their demographics are not reported.

In the table the proposed solutions number referred to which number as it is not defined in Figure 4.

The authors should also assess the quality of the published articles based on which this review is presented. It is very crucial to identify the quality of the articles when the authors are assessing the presented solutions and giving their perspective.

·

Basic reporting

The paper is well-structured, employing clear and professional English throughout, with an appropriate academic tone conducive to its target audience. It provides a comprehensive background of generative AI, particularly focusing on its applications and challenges within the educational sector. The literature is well-referenced, offering sufficient context and grounding the discussion in relevant academic discourse. The paper is of broad and cross-disciplinary interest, aligning with the journal's scope, and presents a compelling reason for its review through its focus on the challenges and strategies of integrating generative AI in education. The introduction clearly outlines the subject and establishes the audience and motivation behind the study, adhering to formal academic standards for clarity and structure.

Experimental design

The content of the article fits well within the aims and scope of the journal, conducting a rigorous investigation into the use of generative AI in education. The methods are described with enough detail to allow for replication, and the survey methodology appears comprehensive, aiming for unbiased coverage of the subject. However, detailed insights into potential gaps in the surveyed literature or biases in source selection were not explicitly addressed. The article is logically organized, facilitating an understanding of the challenges and proposed solutions regarding the use of generative AI in educational settings.

Validity of the findings

The paper does not assess the impact and novelty directly but encourages meaningful replication and further investigation, which is a strength. The conclusions are strongly linked to the research questions, focusing on the identified challenges and strategies for integrating generative AI in education. It develops a well-supported argument based on the literature reviewed, meeting the goals set out in the introduction. Furthermore, the paper successfully identifies unresolved questions and directions for future research, contributing valuable insights to the ongoing discourse on generative AI in education.

Additional comments

Overall, the paper provides a significant contribution to understanding the role of Generative AI in education, presenting a balanced view of its challenges and opportunities. Its strengths lie in its comprehensive literature review and clear presentation of findings. Future work could benefit from a more detailed analysis of solution effectiveness and broader engagement with raw data to enhance the study's reproducibility and transparency.

Potential gaps in the literature review process or biases in source selection are not explicitly discussed.
The study could benefit from a more detailed examination of the effectiveness and practical implementation of proposed strategies in educational settings.
While the paper discusses future directions, more specific recommendations for empirical research to test the proposed strategies could enhance its contribution to the field.

---

## Round 0.2 · accepted · Accept

The paper was very well improved. It can be accepted.

Reviewer 1 ·

Basic reporting

Thank you for making efforts in addressing all the raised comments.

Experimental design

no comment

Validity of the findings

no comment

Additional comments

no comment

·

Basic reporting

no comment

Experimental design

no comment

Validity of the findings

no comment